# Health and wellbeing of indigenous older adults living in the tea gardens of Bangladesh

Farah Naz Rahman[ID][1]*, Hafiz T. A. Khan[ID][2], Md. Jahangir Hossain[1], Anthony Obinna Iwuagwu[3]

1 Centre for Injury Prevention and Research Bangladesh (CIPRB), Dhaka, Bangladesh, 2 Health Promotion & Public Health, College of Nursing, Midwifery and Healthcare, University of West London, London, United Kingdom, 3 Department of Social Work, University of Nigeria, Nsukka, Nigeria

* farah.naz@ciprb.org

## Abstract

### Background

There are currently 1.5 million indigenous people in Bangladesh, constituting 1.8% of the total population and representing one of the country's most deprived communities. This study explores the health status and quality of life along with their determinants among indigenous older people in Bangladesh in order to fill the knowledge and evidence gap on this topic.

### Methods

A mixed-methods approach was deployed in October 2019 in the Sylhet division of Bangladesh which involved a cross-sectional survey among 400 indigenous older adults (200 males, 200 females) from 8 tea gardens using a pre-tested semi-structured questionnaire. Ten in-depth interviews were also conducted with providers of the tea garden health facilities. Descriptive analysis, multiple logistic and multi-nominal linear regression were performed to explore associated factors around health and quality of life.

### Results

Of the total respondents, the majority (79.5%) had chronic diseases, with visual difficulty being predominant (74%) among the conditions. Almost all (94%) of the respondents experienced delays in receiving treatment and poverty was identified by most (85%) as the primary cause of those delays. Extreme age, being male, living alone and low family income were significantly associated with suffering from chronic conditions. Furthermore, having a chronic condition and extreme age were found to be significantly associated with a low quality of life. Health service providers identified lack of logistical support in the health facilities, the economic crisis and lack of awareness as the major causes of poor health status and poor health seeking behaviour of the indigenous older adults.

### Conclusion

Indigenous older men in extreme old age are more vulnerable to adverse health conditions and poor quality of life. Health literacy and health seeking behaviour is poor among

**Data Availability Statement:** All relevant data are within the manuscript and its Supporting Information files.

**Funding:** Centre for Injury Prevention and Research Bangladesh (CIPRB) funded this study.

Website of the organization: https://www.ciprb.org/
The funders had no role in study design, data collection and analysis, decision to publish, or preparation of the manuscript.

indigenous older adults generally and there is a huge gap in the health services and social supports available to them.

## Introduction

In 2019, more than 700 million people worldwide were estimated to be aged 65 years and above [1], forming approximately 9% of the world's total population. The old age (60+) population has tripled in 50 years since 1950, and it is expected to triple again by 2050 [2]. While many high income countries (HIC) have already started to experience the impact of population aging, low and middle-income countries (LMIC), particularly in the South East Asian region, are considered to be the new epicentre with one of the fastest-growing aging populations [1]. According to the UN report, the proportion of the population aged 65 years or above nearly doubled in the South East Asian region from 6% in 1990 to 11% in 2019 [1]. Bangladesh shares a similar scenario, seeing a 25 year increase in life expectancy over a 50 year period from 1970 to 2019 [3]. It is further projected that about 40% of its total population will be aged 60 years or over by 2050 [4].

The emerging aging population has significant financial and healthcare implications. For countries like Bangladesh, where public transfers are relatively small, there is greater strain on individuals and families to support their consumption during old age [1]. The growing number of the older population, with comorbidities and deteriorating functional status in many cases, demands substantial healthcare services that would consume a large portion of healthcare costs at national and family level [5]. Despite the fast growing trend, there is still a lack of focus in Bangladesh in its policies and programmes for dealing with older care. In this situation, where mainstream older adults in Bangladesh are struggling to access their rights, it means that minority groups, such as the indigenous peoples, are likely to experience an even higher degree of vulnerability and lack of access.

Indigenous peoples are distinctive cultural societies and communities that were the first inhabitants of a country or geographical region before subsequently being dominated by newcomers or settlers [6]. This group of people are often the subject of discrimination and inequality compared to mainstream settlers. Approximately 15% of the world's extreme poor is made up of indigenous groups that also face frequent impediments in accessing basic resources, services, justice and rights [7]. Bangladesh currently has 1.5 million indigenous people that constitutes about 1.8% of the country's total population [8]. Although some community representatives like the Bangladesh Indigenous People Forum (BIPF) claim that the actual number is around 5 million [9]. Despite the considerable number, they are still one of the most deprived, neglected and discriminated groups in Bangladesh in all sectors covering health, education, the economy, and political rights [8,9]. The indigenous population of Bangladesh is in a long-term dispute with the government over land rights, resulting in the militarization of the majority of areas where they live [9]. Furthermore, the differences in language and culture from the mainstream population greatly hinders their integration into the education and employment sectors [10]. According to data from the indigenous navigator, poverty among Bangladesh's indigenous population is three to four times higher than the national average with very limited coverage from the community social support system whether provided by the public or private sectors [10].

Several research studies have been conducted into the HICs exploring the health status, quality of life, and support system of indigenous older adults [11,12]. A couple of studies that compared indigenous and non-indigenous Australians found that the health status and health

utilization among the indigenous group was significantly worse than among the settlers [13,14]. Another two studies conducted in Canada and Chile revealed that the indigenous status is significantly associated with poorer quality of life [15,16]. Although having demonstrated disparities, research-based evidence on the healthcare needs and quality of life of indigenous populations of LMICs, particularly in the older age group, is very limited. As the governance and social support system in the LMICs differs from the HICs, and there are significant cultural differences between the indigenous communities around the world, evidence from the HICs may not be appropriate for the development of policies and interventions for the LMIC indigenous communities.

In Bangladesh, the few studies that did involve the indigenous group focused mainly on maternal and child health within the communities [17–19]. Studies focusing on the health situation of older indigenous people are rare, and the one study that explored the self-assessed health status of older teagarden workers found a high prevalence of multimorbidities among them [20]. However, the healthcare needs, health status, and its relationship with quality of life of indigenous older adults in Bangladesh are still unknown and unexplored, which signifies the need for research specific to this community. It is essential to generate evidence regarding the challenges and needs of these vulnerable communities in order to strategize tailored policies and programmes for the marginalized groups while addressing the developmental agendas of the older population.

This study, therefore, aims to explore the health status and quality of life and to define its determinants among the indigenous older adults residing in tea gardens of Bangladesh.

## Methodology

### Study setting

This study was conducted in the Sylhet division of Bangladesh that is situated in the northeastern part of the country and is one of the places where the indigenous or ethnic minority population predominantly resides [21]. Indigenous communities in this area primarily live in the tea gardens and work in the tea estates [22].

### Sample and sampling technique

Around 80% of the older population of Bangladesh are suffering from some type of chronic condition [23–26]. Information on the health status of the indigenous population is scarce, but it is widely acknowledged that they are deprived of basic human rights [21,27] and have very limited access to health facilities compared to the general citizens [28]. Therefore, this study assumed that the proportion of those in the indigenous older population having a good health status and quality of life would stand at 15%. Taking this 15% prevalence plus 80% power and 95% CI, the estimated sample size is 196 and then multiplying with the gender strata (196*2) comes to 392 that is then rounded up to 400 considering a 90% response rate.

The Sylhet Division is divided into four districts and a further 35 sub-districts or upazila. Two upazila, Sreemangal and Kamalganj were purposively selected for this study as the Centre for Injury Prevention and Research Bangladesh (CIPRB) has been working in these areas since 2016 maintaining a good relationship with both the tea garden management authorities and people in the community over the years. Eight Tea Gardens-Amrail, Rajghat, Khejuri, Khaichara, Mirtinga, Shamshernagar, Alinagar and Patrokhola were randomly selected from two upazila (4 from each) and a further 5 labour blocks, locally termed as 'Panchayet' were selected randomly from each tea garden. From each 'panchayet', 10 indigenous older adults aged 60 years or above (5 males, 5 females) were randomly selected and 400 indigenous older people (200 males, 200 females) were then included in this study.

**Table 1. Description of socio-demographic and health condition variables used in analysis with their percentage in total.** (N = 400).

| Characteristics of Respondents | Measurement of variables | N | Percentage (%) Total |
|---|---|---|---|
| Age | Age 60 to 69 years | 226 | 56.5 |
| | Age 70 to 79 years | 132 | 33 |
| | Age 80+ years | 42 | 10.5 |
| Gender | Male | 200 | 50 |
| | Female | 200 | 50 |
| Marital Status | Currently Married | 242 | 60.5 |
| | Single/Divorced/Widow | 158 | 39.5 |
| Income (weekly) | 500 and below | 323 | 80.8 |
| | 501 + | 77 | 19.2 |
| Education | Illiterate | 276 | 69 |
| | Primary | 92 | 23 |
| | Above primary | 32 | 8 |
| Suffering from chronic disease | Yes | 318 | 79.5 |
| | No | 82 | 20.5 |

## Socio-demographic characteristics of the respondents

The socio-demographic characteristics of the respondents are presented in Table 1 and Fig 1.

Of the total respondents, more than half fall into the age category of 60–69 years. In this age group, the proportion of males and females is almost the same but in the extreme age group (80+ years), older females constituted two-thirds of the respondents. A majority (60.5%) of respondents lived with their partners with older women being the predominant group among those that were single, divorced or widowed. A large number (69%) of respondents did not have any formal education with the rate of illiteracy among older women almost double that of older men. Most of the respondents (80.8%) had a weekly income of less than 500 Bangladeshi Taka (BDT) (equivalent to USD 6).

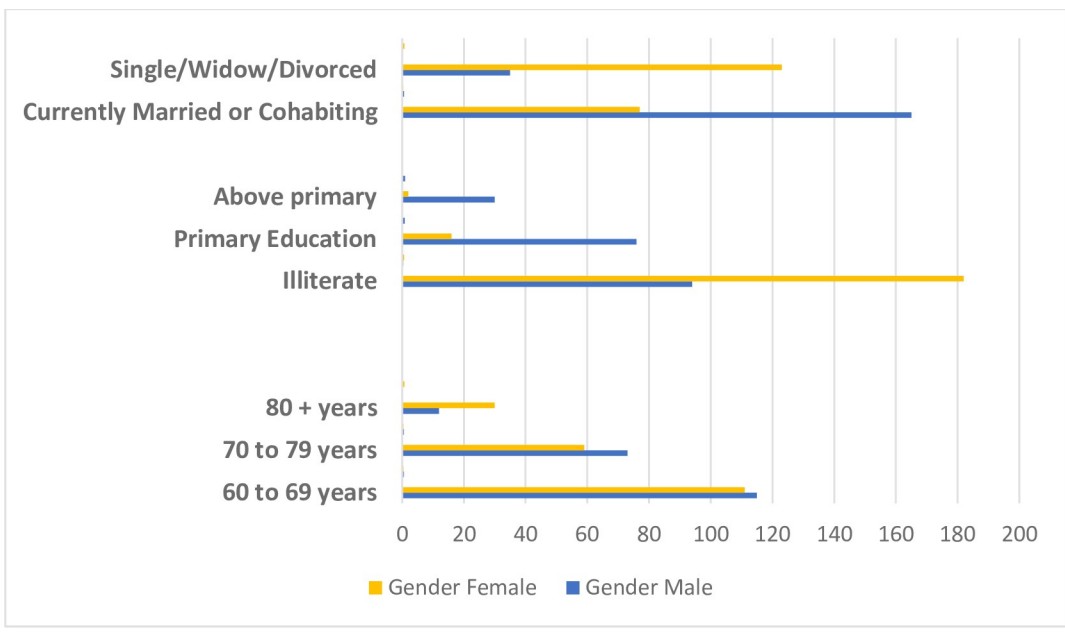

**Fig 1. Distribution of age, education and marital status of indigenous older adults by gender.**

## Study design and data collection

The study deployed a mixed methods approach which included a cross-sectional survey and ten in-depth interviews. The cross-sectional survey was carried out among the 400 indigenous older people using a pre-tested semi-structured questionnaire. Subsequently, ten in-depth interviews (IDI) were conducted with health care providers from 8 tea gardens. IDI participants were selected from health service providers of different educational and social background as part of the data triangulation process. The respondent were 3 registered medical doctors, 3 compounders, 3 paramedics, and one community healthcare worker. Each interview lasted from 30 to 60 minutes.

**Ethics statement.** Informed written consent was taken from all the respondents. Anonymity and confidentiality of the respondents were maintained throughout the data collection, handling, and analysis process. Ethical approval was taken from Institutional Ethical Review Board of CIPRB.

## Instrument

A 41-item questionnaire was used to collect information on sociodemographic characteristics, health status, health-seeking behavior, and quality of life of the indigenous older adults. The questionnaire can be found in S1 Appendix.

To explore the health status of the participants, information was obtained on presence of chronic condition, severity of chronic condition, and whether the respondent face difficulties in performing daily life activities. All the information of health status and functional ability was based on participant's self-reported or perceived health condition. To determine the presence of a chronic condition or disease, the respondents were asked a close-ended question about whether they had been suffering from any health related condition over the previous 6 months. The responses were recorded as 'Yes/No'. They were further asked to share about their condition and it was categorized under eight domains on which older people have prevailing morbidity. The domains are- visual, hearing, locomotor, cardiac, respiratory, gastrointestinal, nutrition, and mental health. Participants who reported health problem in more than one domains were considered as having multimorbidities. Their perception on the severity of their chronic condition was further obtained through a 3 level Likert scale categorized as- little difficulty, average difficulty, and severe difficulty. For example, participants who shared having little difficulty in seeing distant object was recorded as having little difficulty in visual morbidity domain. In addition, participants were asked whether they feel that their physical condition hinders them from performing out major everyday activities, which are: household work, climbing stairs, body movements (bending, kneeling, stooping) and walking a few yards. The responses were recorded as 'Yes/No'. Furthermore, those who reported having difficulties in performing daily life activities, were asked to rate their limitation in a 3 level Likert scale categorized as- limited a lot, moderate limitation, limited a little. Additionally, information on whether they seek treatment for their chronic condition, from where they seek/receive treatment, delay between onset of symptom and receiving treatment along with causes of the delay, and adherence to treatment were obtained to explore the health-seeking behavior of the respondents.

In order to measure their quality of life, 6 Likert scale questions were used with each question bearing equal weight and having five response values from 1 being 'strongly disagree' to 5 being 'strongly agree'. Responses from each of the questions were added to develop a quality of life score where a higher score suggested a high quality of life. In this way, the quality of life score ranges from 6 to 30. This questionnaire was adapted from a six-item short version of the WHOQOL-OLD module which evaluates the quality of life of an older adult under six

domains- sensory abilities, autonomy, death and dying, social participation, intimacy, and past, present and future activities [29]. The Cronbach's alpha of the quality of life measurement scale was 0.66. The list of questions can be found at the section-4 of the questionnaire attached as S1 Appendix.

A literature guided IDI guideline was used to gather information on the health status and health-seeking behaviours of the indigenous older adults from the healthcare providers of the tea-gardens. The IDI guideline can be found in S2 Appendix.

## Statistical analysis

Quantitative data were analysed using SPSS software v24. A Chi-square test was carried out to identify any association between demographic and health variables and a further logistic and linear regression helped to determine the factors that played a role in promoting a good health status and quality of life. Multilinearity, homoscedasticity and normal distribution tests were conducted to determine whether the data met the assumptions of multiple linear regression. The outcomes of Variation Inflation Factors (VIF) and tolerance score revealed that the collinearity assumption was met. The histogram and scatterplot of standardised residuals confirmed that the data met the assumptions of normality, homogeneity, and linearity. The statistical significant level of findings was considered as p value <0.05.

Qualitative data collected from healthcare providers were transcribed from audio-recordings and translated into English and thematic analysis was performed on the interview transcripts with an open coding approach adopted in order to explore the themes. The themes of this study aimed at capturing the common concepts of the indigenous older adults that were clustered around a core idea of health status and health-seeking behaviour. The qualitative data were used to supplement the quantitative findings and triangulated interpretations were discussed to demonstrate a comprehensive health scenario of the indigenous older adults.

## Results

### Health status of the indigenous older adults

A majority of respondents (80%) were suffering from some type of chronic disease.

Fig 2 portrays the distribution and severity of chronic conditions among the respondents. Visual difficulties were found to be predominant (74%) followed by locomotion difficulties (49%) and gastrointestinal problems (41%) with a significant number of older adults suffering from hearing difficulties and malnutrition. Of the respondents with visual difficulty, 30.5% reported the problem as severe with more than 10% also complaining of severe difficulties with hearing and gastrointestinal-related problems.

Furthermore, Fig 3 presents the distribution of multimorbidities among the indigenous older adults across male and female. Almost all of the respondents (90%) were suffering from multi-morbidities and more than 20% had morbidities in five or more areas. Older women have higher proportion of morbidities in 2–5 domains when compared with men. The prevalence of multi-morbidity in more than five domains was higher among older men.

A large number (69.6%) of respondents reported difficulty in carrying out household tasks.

Fig 4 presents the severity of the limitation on various daily activities of indigenous older adults. About one-fourth of the respondents complained of severe limitations in their abilities to climb steps or in walking a few steps or moving their bodies such as bending, kneeling or stooping. Additionally, more than 40% of the respondents had average or moderate limitation in all domains covering their daily activities.

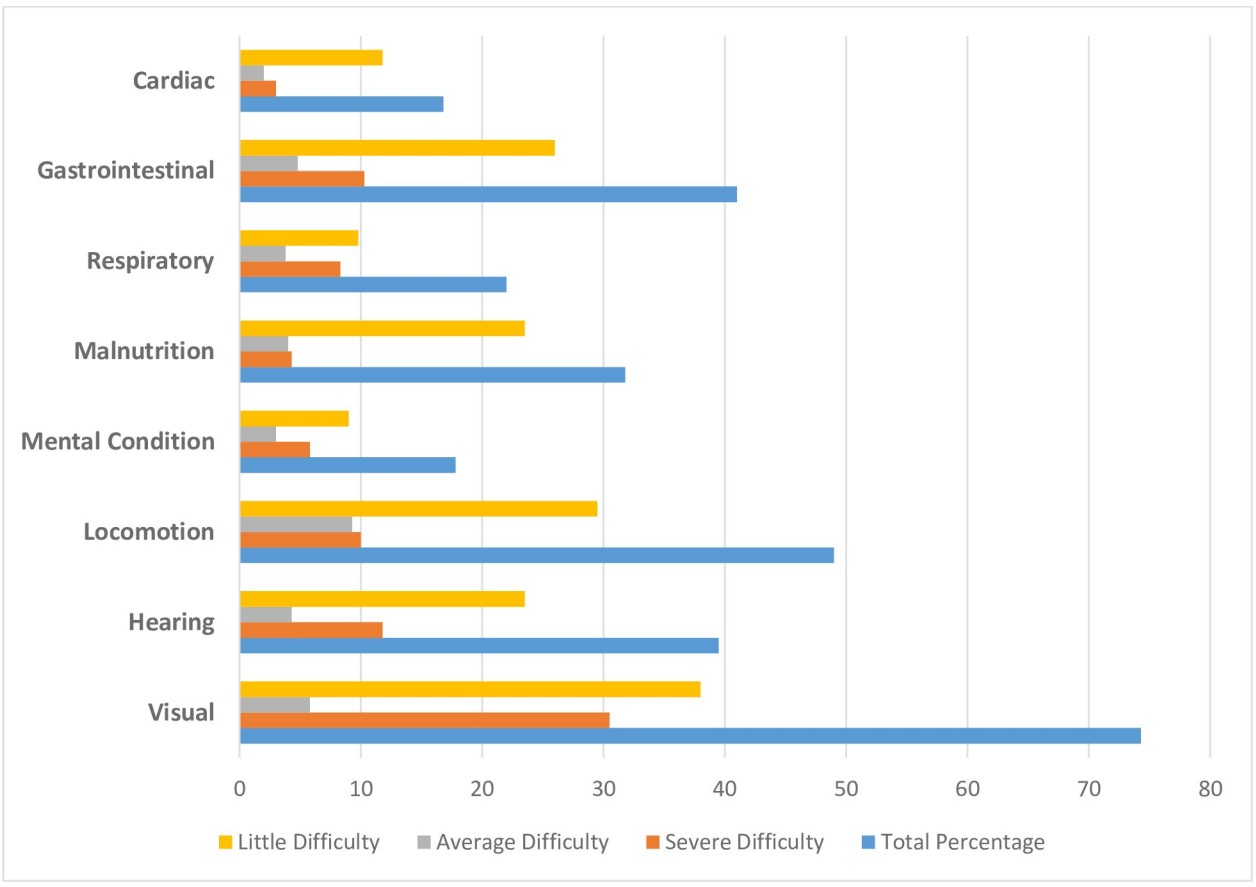

**Fig 2. Distribution of chronic conditions of indigenous older adults with their level of severity.**

## Health seeking behaviour of the indigenous older adults

More than one-third of respondents did not receive any treatment for their chronic conditions or morbidities. Among those who did seek treatment, 5.3% took services from unregistered facilities such as quacks, homeopathy, dispensaries and kabiraji, for example.

The facilities where indigenous older adults used to seek treatment for their health conditions is demonstrated on Fig 5. Most of the older adults (40%) would go to the tea garden health facilities for treatment and of those that did this, only one-third (35.8%) were able to receive treatment from a medical doctor whereas others received treatment from paramedics and field health workers. Almost all respondents (94%) delayed seeking treatment from the onset of disease with poverty identified by a majority (85%) as the cause of such delays followed by lack of knowledge on the consequences or severity of the particular disease.

## Determinants of health and quality of life of indigenous older adults

The relationship between socio-demographic characteristics of the respondents and the predisposition to suffering from chronic disease was analyzed by multivariate binary logistic regression. The dependent variable in this analysis is 'persons suffering from chronic disease' coded so that 0 = Not suffering and 1 = Suffering from disease. The independent or predictor variables were age, sex, marital status, income, and education. The reference category for each predictor variable is presented in the first row under individual variable. The statistically significant results are marked with star (*).

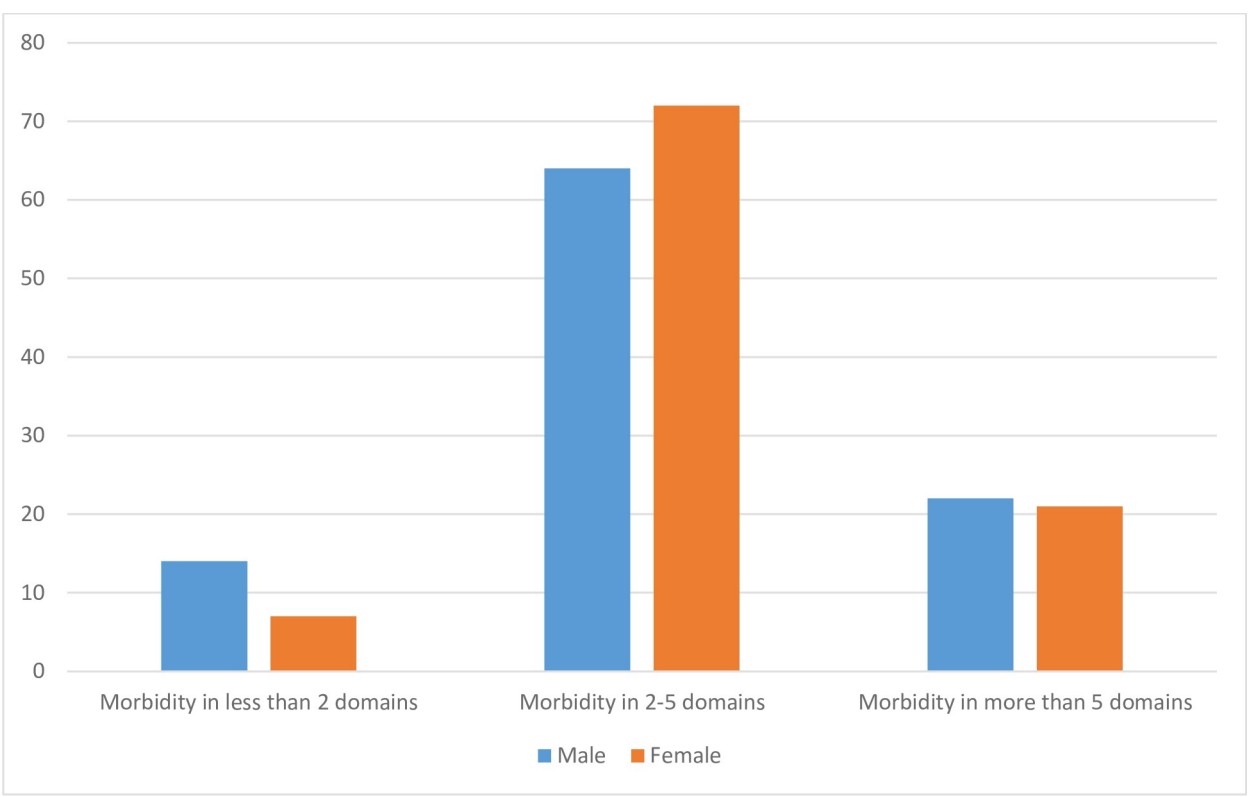

**Fig 3. Distribution of multi-morbidity among indigenous older adults across gender.**

The multiple logistic regression shown in Table 2 reveals that older indigenous men have two times higher chance of suffering from chronic disease than older females. Marital status of the respondents is also found to be significantly linked with suffering from chronic disease with the odds of this happening being 11 times higher among indigenous older adults that are currently living alone than among those that are cohabiting. The economic condition of the indigenous older population is also significantly associated with their chronic conditions. Those that have a weekly income of less than 500 BDT are 2.3 times more likely to suffer from chronic disease than older people that have a higher weekly income. Furthermore, the results revealed that indigenous older adults aged between 60 to 69 years, have 2.8 times higher odds of suffering from chronic diseases than older adults who are aged 80 years and above. Education was not found to have a significant association with the chronic conditions of indigenous older adults.

Furthermore, Multiple linear regression was run to predict quality of life from age, gender, income, education, marital status, and condition of chronic disease. The relationship of quality of life with individual predictor variable was explored in the unadjusted model. Additionally, in the adjusted model, the effect of all other covariates were controlled while demonstrating the predictive value of an independent variable on quality of life of the older adults. Among the independent variables, only age was continuous variable measured in years. The categorical independent variables consisted of gender (male vs female), marital status (currently married/ cohabitating vs single/divorced/widow), income (>500 BDT weekly vs ≤ 500 BDT weekly), education (no literacy, primary [equivalent to 5 years of schooling], above primary [equivalent to >5-years of schooling], and suffering from chronic disease (yes vs no). The reference

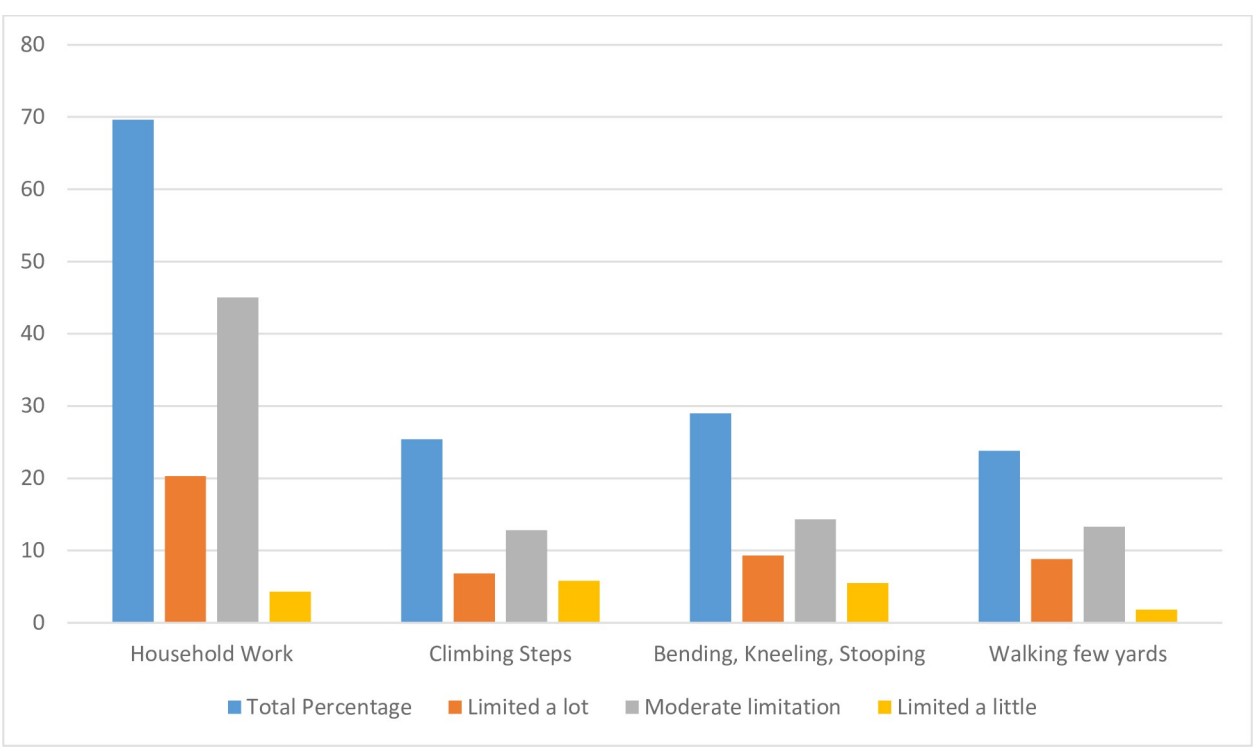

**Fig 4. Limitation on daily activities among indigenous older adults with severity.**

category used in multiple linear regression for gender, marital status, income, education, and chronic disease condition was consecutively female, currently married, weekly >500 BDT, above primary, and yes = suffering from chronic disease. The statistically significant values were marked as (*) which is equivalent to p value <0.05 and (**) that is equivalent to p value <0.01.

Age and status of chronic conditions were significantly associated with quality of life, both in adjusted and unadjusted models as shown in Table 3 [F (7, 392) = 6.756, $p<0.00$ and $R^2$ = .108]. Age was found to have an inverse association with quality of life of the indigenous older adults where there was a 0.5-unit decrease in quality of life with one-year increase in age. When compared to the indigenous older adults those are free from any chronic conditions, the quality of life for older adults suffering from chronic diseases decreases by 2 units. Income and marital status are found to affect quality of life significantly in the unadjusted model. Older adults that have a weekly income of less than 500 BDT have a decreased quality of life by 1.15 units than those that have a weekly income higher than 500 BDT. Similarly, indigenous older people that are living alone have a decreased quality of life by 1.25 units compared to older adults that are currently married or cohabiting. Gender was found to have no significant effect on the quality of life of the indigenous older population.

The healthcare workers of tea garden health facilities have shared their views and recommendations on the health status and health seeking behaviour of the indigenous older population, along with its associated challenges. The themes emerged from the qualitative analysis were common health problem of the indigenous older adults, practice of preventive health behavior, factors associated with health status and health-seeking behavior, inequality in health condition, challenges of older people in maintaining good health, and recommendation for improving health condition and health services to indigenous older adults.

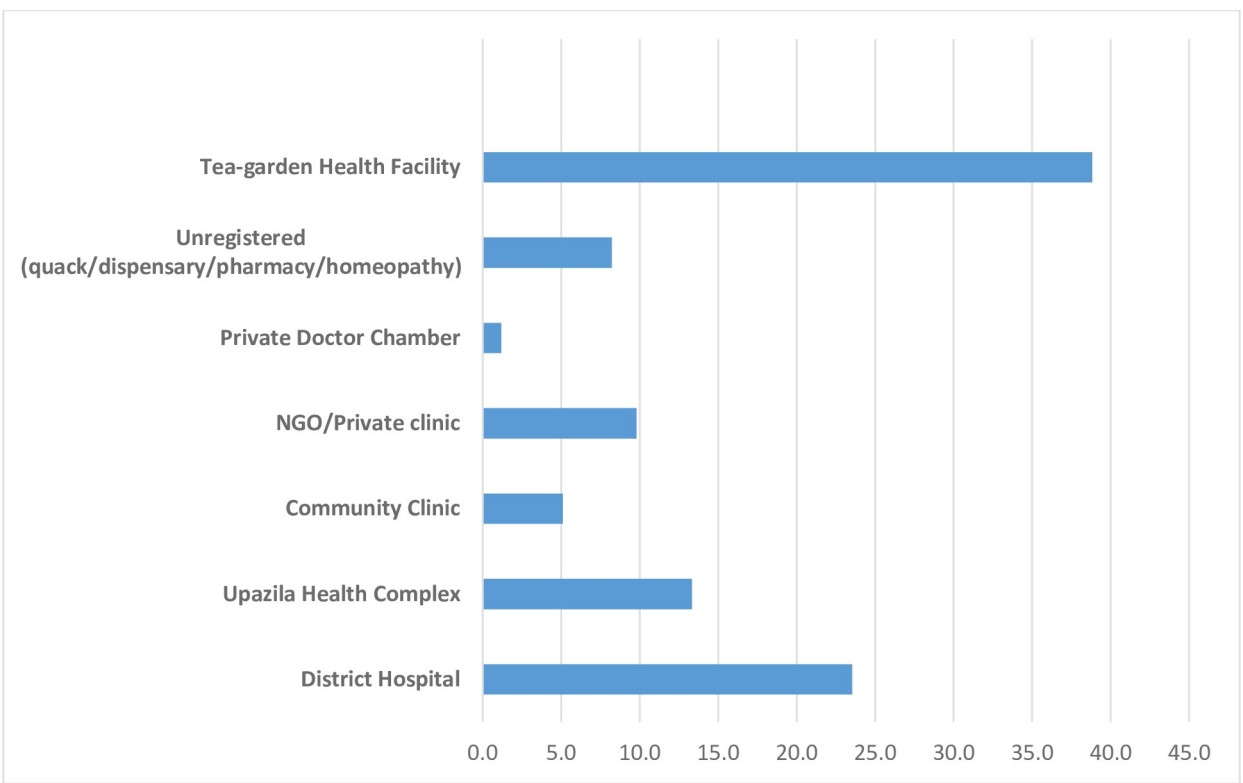

**Fig 5. Facilities from where indigenous older adults received treatment.**

**Table 2. The results of multiple logistic regression analysis on predicting persons suffering from chronic disease based on age, gender, marital status, income, and education.**

| Independent Variables | B | S.E | Sig. | OR | 95% CI |
|---|---|---|---|---|---|
| **Age** | | | | | |
| 80+ years | 0.00 | | | 1.00 | |
| 70–79 years | .800 | .494 | .105 | 2.226 | .845–5.863 |
| 60–69 years | 1.062* | .478 | .026 | 2.891 | 1.134–7.372 |
| **Gender** | | | | | |
| Female | 0.00 | | | 1.00 | |
| Male | .776** | .320 | .015 | 2.172 | 1.160–4.066 |
| **Marital Status** | | | | | |
| Currently married/Cohabiting | 0.00 | | | 1.00 | |
| Single/Widow/Divorced | 2.461** | .421 | .000 | 11.722 | 5.137–26.745 |
| **Income(weekly)** | | | | | |
| 501 + | 0.00 | | | 1.00 | |
| 500 and below | .836* | .320 | .009 | 2.307 | 1.232–4.319 |
| **Education** | | | | | |
| No literacy | 0.00 | | | 1.00 | |
| Primary Education | .312 | .353 | .376 | 1.366 | .685–2.727 |
| Above primary | .467 | .564 | .407 | 1.596 | .528–4.822 |

**Table 3. The results of unadjusted and adjusted models of linear regression analysis predicting quality of life of indigenous people by age, gender, marital status, income education, literacy, and presence of chronic disease.**

| Variables | Unadjusted Model | | | Adjusted Model | | |
|---|---|---|---|---|---|---|
| | B Coefficient | Sig | 95% CI | B Coefficient | Sig | 95% CI (Lower, Upper) |
| Age (In years) | -.068 | .003 | -.113, -.023 | -.052* | .024 | -.098, -.007 |
| Male | .450 | .189 | -.223, 1.123 | -.202 | .626 | -1.017, -.613 |
| Single/Divorced/Widow | -1.256** | .000 | -1.935, -.578 | -.475 | .244 | -1.276, .328 |
| 500 and below weekly | -1.153* | .008 | -2.000, -.305 | -.350 | .429 | -1.220, -.520 |
| No literacy | -.678 | .067 | -1.404, .048 | -.946 | .149 | -2.233, .343 |
| Primary | .471 | .248 | -.329, 1.270 | -.334 | .620 | -1.659, .991 |
| Chronic Disease (Yes) | -2.284** | .000 | -3.088, -1.480 | -2.069** | .000 | -2.922, -1.216 |

## Health status of the older population in tea gardens

According to the respondents, common health problems that occur among the older population in the tea gardens include cataracts, high blood pressure, stroke, musculoskeletal pain (predominantly back pain), tuberculosis, diarrhoea, and malnutrition. Fever, fractures, and headache are among the most common symptoms presented by older adults in health facilities. A few older adults also suffer from different forms of cancer, with prostate and lung cancers predominating among them.

The respondents further reflected on the various causes that are responsible for the health situation of the indigenous older adults. One respondent said:

> *"Due to the use of drinking water and well water in daily life, the tendency of diarrhoea in the garden is alarming. Sometimes it is epidemic. Unhealthy living leads them to various diseases."*—(IDI-9, Tea garden dispensary).

The respondents further shared that many older people in the tea gardens had malnutrition due to a lack of sufficient nutritious food. They identified economic condition of the older adults as the primary cause of malnutrition. A respondent shared his views:

> *"Their income is very poor. They earn only BDT 102 per day despite working from sunrise to sunset. It is obviously not sufficient for them to maintain the bare minimum of a healthy life."*—(IDI-5, Community clinic).

The IDI respondents further shared that the rate of alcohol consumption is also high among older people. They spend a large part of their income to buy alcohol, which is a tradition in the tea gardens that started nearly 100 years ago.

Furthermore, the respondents shed light into the presence of substantial gender disparity and cultural inequality among the indigenous older community. Women suffer from a variety of diseases and because they are not particularly empowered, they are neglected, deprived and disadvantaged in the family life as well as in the community. Most of the older people are from traditionally lower caste of Hindu religion and therefore, are bound by many religious beliefs regarding the health service that caused more suffering for them than for other groups. Additionally, there is a prevalent habit of taking tea mixed with salt among the older adults which potentially contribute to the prevailing cardiovascular diseases. Moreover, older people have poor hygienic practices, such as not washing hands after going to the toilet, that make contracting diseases more likely.

### Health seeking behaviour of the older population in tea gardens

The health seeking behaviours of the older population living in tea gardens was considered very poor by all the respondents, with ignorance and lack of awareness being the main identified cause.

> *"When the older people face any injury; They ask for X-ray and believe that they will be fine after performing it. They considered X-ray as a treatment for injury. I try to convince them, but it is very difficult to change their believe. That's how much ignorant they are about their medical conditions"*- (IDI-5, Teagarden hospital).

Older people tend to be neglected by other family members and by the tea garden authorities as many of them have difficulty in performing productive works. Many older people are addicted to alcohol, which is not prohibited in the tea garden area, and will often delay visiting a dispensary to get treatment for a disease until quite late in its progress, and even when they do visit, adherence to medication is not adequate.

To elaborate further about the lack of awareness among older adults, the respondents shared that a substantial number of older people are depended on the rural kaviraj and traditional healers. After having an illness, the older adults tend to neglect it and seek care from Kaviraj, Puruhits and Ojhas rather than going directly to the registered health facilities. This causes a significant delay in beginning proper care, as they usually arrive at registered health facilities after their condition become serious. Respondents also expressed their concern that many women suffer from cervical cancer and fistula, but are not able to move outside the tea gardens for social reasons, and thus it is difficult to ensure that a referral to an outside hospital is pursued.

### Challenges faced by the older population within tea gardens

Several challenges were identified by healthcare providers that older people commonly face in the tea gardens. The primary cause is the inability to afford the costs necessary to receive adequate health care management. Since only registered tea-garden workers are entitled to free treatment, older adults face significant difficulties in obtaining adequate health services for their illness after retirement.

> *"Tea-garden workers at their old age hardly have any savings. It is very difficult for them to bear their medical expenses without any external support."*–(IDI-8, Tea-garden dispensary).

The healthcare workers further shared that there is lack of logistic support and technical expertise in the teagarden health facilities for addressing the complex chronic conditions of the older people. Furthermore, older people are often unable to go to other facilities following referral due to lack of accompanying persons and economic affordability. Although the tea garden authorities are responsible for providing accommodation, safe water, sanitation, medical services and educational facilities for the workers, there is substantial lack in basic services available to them.

### Recommendations of healthcare providers on the healthcare of older people within tea-gardens

An awareness raising programme among older people in the tea gardens that focuses on health education and personal hygiene was felt to be essential by Healthcare Providers (HCPs) that took part in the interviews for this study. They also felt that medicines such as calcium,

vitamins, antibiotics, antihistamines, analgesics and beta-blockers should be made available in the tea garden dispensaries. In addition, the HCP recommended that health facilities should have health care staff trained on elderly care for better diagnosis and management of the older adults. Moreover, they advised that the use of alcohol is also need to be prohibited within the tea garden areas.

> *"The older people are not aware about their health issues. Health and personal hygiene related awareness meeting and video documentary may motivate them to change their believe and practices. Available medicine and referral support also need to be ensured in the teagarden dispensaries."*- (IDI-10, Tea garden dispensaries).

Along with the awareness building, the respondents emphasized on developing a proper referral system for advanced management of the critically ill older people. They also felt the need for the local government and the teagarden authorities to establish an easily accessible and affordable transport system so that older adults could reach out to outside health facilities without delay. All the HCPs believed that economic condition of the older indigenous community needs to be improved to ensure their accessibility and affordability to adequate healthcare services. This can further lead to a healthier lifestyle by allowing them to consume nutritious food and improve their sanitation and hygiene behaviour. In addition, health workers indicated that the health and social issues of the older indigenous community should come into the focus of policy and welfare programs carried out by the government, the tea garden authorities and other key stakeholders, which are currently minimal or non-existent. One of the respondent expressed their opinion:

> *"Many organizations, both government and non-government, often come here and carry out a range of welfare activities, such as health campaigns, micro-credit schemes, women's empowerment programs, etc., but none of these are aimed at older people. They are frequently overlooked for services provided by the teagarden authorities and other stakeholders."*- (IDI-4, Tea garden Hospital).

## Discussion

This study explores the impact of socio-demographic factors on the health status and quality of life of the indigenous older population in Bangladesh. The majority of older adults in tea gardens live under very poor socio-economic conditions with a low weekly income of less than 500 BDT. The International labour Organization's report on occupation and incomes of the indigenous population also finds that their average household annual income is 20,000 BDT that is less than 500 BDT a week [30]. Many of them are illiterate that is consistent with the findings of the Chittagong Hill Tracks (CHT) indigenous study where the average tribal population illiteracy rate was 63.5%, that is also substantially higher than the non-tribal population [31]. As the life expectancy of females is higher than males in Bangladesh [32], it is understandable that older females make up the largest number in the extreme age group. It also explains why older women are living alone as traditionally the age gap at marriage between men and women is 10–12 years.

This study found a high prevalence of chronic disease and multimorbidity among the respondents. Similar result is shared by Hossain et al' study on teagarden older population [20]. This is also consistent with Kabir et al's study that found 95% older people living in both rural and urban areas in Bangladesh had health-related problems [33]. This study further reveals that 90% of the indigenous older women suffers from any kind of morbidity, with more than two-third of them having morbidities in 2–5 domains. Kabir et al also reported that

80% of older women had morbidities in four or more domains [33]. Further to this, a large scale ageing survey in Bangladesh showed that 97% of the respondents had some sort of health problem, with significantly more women reporting such problems [34]. Moreover, in this ageing survey, 63.5% of the respondents were suffering from chronic conditions [34], whereas in this current study, chronic disease was present among 79% of the indigenous older population. The proportion of multimorbidity among the indigenous older adults were also found higher when compared with Khanam et al's cross-sectional survey that found multi-morbidities among 53.8% of the rural older population [35]. Difficulties with vision was a predominant condition among the indigenous older group that has also been identified as the predominant issue among the Bengali older population in Kalam et al's study [34]. The HCPs also mentioned about prevalence of cataract among the indigenous older adults in their interviews.

Almost all (90%) the respondents were suffering from any kind of chronic health problem and a large number (70%) of them reported of having difficulties with performing household tasks. This indicates a possibility that the chronic condition might have an influence in limiting their daily activities, which can be investigated with further studies. This finding is consistent with another performance-based evaluation study of the older population of Bangladesh where more than half of the participants reported having difficulty with performing one or more tasks included in a performance test [25]. This evidence indicates that chronic conditions and multi-morbidities are prevalent among the older population generally in Bangladesh but are higher among indigenous older people. The HCPs of local health facilities identified unhealthy lifestyle of indigenous older adults including high rate of alcohol consumption, lack of nutritious food on diet, and poor hygiene behaviour as the important associated factors for high prevalence of morbidity. HCPs further shared that older women suffer more from these conditions due to their disadvantaged position in the community, which is consistent with the findings of a qualitative study conducted among the tribal people of CHT [18].

Male gender, young old (60–69 years' age), single marital status, and low income were found in this study to be the predictive factors among the indigenous older population in Bangladesh for suffering from chronic diseases. Several studies also found association of income with better survival chances and improved health-seeking behaviours among the older people of rural Bangladesh [25,35,36]. Additionally, poverty was specially emphasized by the HCPs as a predominant cause of adverse health conditions of the indigenous older people. Findings concerning the only other study by Hossain et al. investigating the self-assessed health status of the older tea-garden population showed that female and unemployed older adults are at higher risk of multi-morbidity [20]. The difference in gender as predictive factors between this study and Hossain's study can be explained by the differences in study methods and analysis plan. Furthermore, Khanam et al's study with the rural older population also reported higher prevalence of multimorbidity among the persons who are single [35]. Moreover, the large scale ageing survey mentioned earlier in this paper that involved mainstream older population from six districts, also identified single marital status as significant predictors of the prevalence of illness and the duration of suffering from disease [34]. However, on contrary to this study, the ageing survey found females and older adults aged >70 years to be more vulnerable for illness.

Along with a poor health status, this study found that the health-seeking practices among the indigenous older people was inadequate with more than one-third of them not seeking treatment for their health problems. Likewise, Hosain et al's study showed that 27% of the Bangladeshi rural older people did not seek treatment for their various conditions [37]. Lack of awareness and health education among the indigenous older adults were pointed out by the HCPs as the probable causes of poor health-seeking behaviour. Furthermore, about 5% of both teagarden older adults and CHT ethnic minorities relying on traditional healers for their health issues [38]. Only one-third of the indigenous older people of teagardens received

treatment from qualified medical doctors, the proportion of this is much lower (18%) among the overall indigenous population of CHT [38]. This is probably due to the majority of the indigenous older group in this study seeking treatment at a tea garden health facility that is within their reach. Dependence on traditional healers and unqualified health-practitioners among the indigenous population was also surfaced from the interviews with HCPs in this study and from Ahmed et al's study in CHT [38]. Poverty is the major hindrance for seeking appropriate health care both among the indigenous and mainstream older population [37], which also resonates into the opinion of HCPs. The providers of teagarden health facilities recommended the introduction of tailored welfare activities for older indigenous people including health education, health campaign, and social protection measures. They also shared the need for adequate logistic support and professional skills of staff on dealing with older adult's health problems at the teagarden health facilities.

The health status of the indigenous older population is, therefore, impacted by their socio-demographic conditions. In addition, this study found significant association between low quality of life and extreme age, single marital status, poor economic situation, and suffering from chronic conditions, among the indigenous older population. These findings are consistent with the findings of a population-based survey conducted among the older population of rural Bangladesh where advanced age and a low socio-economic situation were found to be significantly associated with a poor quality of life [39]. A couple of studies by Nilsson et al focusing on quality of life of rural older people also identified living in a multi-family member and being healthy as significant determinants of quality of life [39,40].

## Strengths and limitations

One of the limitations of this study is that it used self-rated health status from respondents that could be subjective and sometimes prone to recall bias. However, to help reduce any bias, respondents were also asked to provide relevant evidence such as prescriptions and medicines for example. A qualitative study involving the indigenous older adults to further explore their quality of life and its predictive factors in order to supplement the quantitative results would have been useful.

Nonetheless, this is probably the only study that attempts to explore the health status, health-seeking behaviour, and quality of life of the indigenous older population residing in the tea gardens in Bangladesh. Other studies regarding ethnic minorities mostly involve the population from CHT and focus mainly on reproductive and child health issues [18,38,41]. This study can serve as an evidence base for further in-depth research and policy interventions. The study implies the need for policies to improve the coverage of the old-age welfare system, including marginalized groups such as indigenous peoples, and to develop tailored health funding and awareness-raising strategies for indigenous older adults.

## Conclusion

The indigenous older population in Bangladesh are vulnerable and have a poor health status and low quality of life. They suffer more due to adverse health conditions than the mainstream Bengali population that ultimately poses a negative impact on their quality of life. The determinants of health and quality of life are similar to those of the general older population and yet, they suffer more because they are deprived of basic human rights and experience endemic poverty.

The older population overall in Bangladesh are in a vulnerable state, but those of the indigenous population are extremely vulnerable and demand special attention. They have been severely neglected and the limited interventions that there have been were targeted at other age-groups. Urgent policy level interventions are therefore required. Poverty is a major cause

of vulnerability and it is recommended that pension benefits are made available to all indigenous older adults. Additionally, awareness program and health campaigns focused on elderly health should be carried out in order to increase health literacy and improve health-seeking behaviour. Since they mostly depend on the tea garden health facilities, proper manpower and technical support should be provided to those facilities. Taking action to promote the good health of the indigenous older population in Bangladesh should be given priority in recognition of its key role for enabling them to lead a better quality of life.

## Supporting information

**S1 Appendix. Instrument used in cross-sectional survey among indigenous older adults residing in tea-gardens of Bangladesh.**
(PDF)

**S2 Appendix. Guidelines for in-depth interviews conducted among healthcare providers of tea-gardens in Bangladesh.**
(PDF)

**S1 Data. Dataset of the cross-sectional survey among indigenous older people in tea-gardens of Bangladesh.**
(SAV)

## Acknowledgments

The study was conducted in October 2019 while first author visited University of West London under fellowship of the Charles Wallace Trust, UK. Authors express their gratitude to Oxford Institute of Population Ageing, University of Oxford for providing academic support to collate literature review of the study.

## Author Contributions

**Conceptualization:** Farah Naz Rahman, Hafiz T. A. Khan.

**Data curation:** Farah Naz Rahman, Hafiz T. A. Khan.

**Formal analysis:** Farah Naz Rahman, Anthony Obinna Iwuagwu.

**Funding acquisition:** Farah Naz Rahman.

**Investigation:** Farah Naz Rahman.

**Methodology:** Farah Naz Rahman, Hafiz T. A. Khan.

**Project administration:** Farah Naz Rahman, Md. Jahangir Hossain.

**Supervision:** Hafiz T. A. Khan.

**Writing – original draft:** Farah Naz Rahman.

**Writing – review & editing:** Farah Naz Rahman, Hafiz T. A. Khan, Md. Jahangir Hossain, Anthony Obinna Iwuagwu.

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
