## [Decision Letter · Decision Letter 0]

30 Nov 2020

PONE-D-20-33976

Health and Well being of Indigenous Older Adults Living in the Tea Gardens of Bangladesh

PLOS ONE

Dear Dr. Rahman,

Thank you for submitting your manuscript to PLOS ONE. After careful consideration, we feel that it has merit but does not fully meet PLOS ONE’s publication criteria as it currently stands. Therefore, we invite you to submit a revised version of the manuscript that addresses the points raised during the review process.

We look forward to receiving your revised manuscript.

Kind regards,

Filipe Prazeres, MD, MSc, Ph.D.

Academic Editor

PLOS ONE

Journal Requirements:

1, Please ensure that your manuscript meets PLOS ONE's style requirements, including those for file naming. The PLOS ONE style templates can be found at

2.Your ethics statement should only appear in the Methods section of your manuscript. If your ethics statement is written in any section besides the Methods, please move it to the Methods section and delete it from any other section. Please ensure that your ethics statement is included in your manuscript, as the ethics statement entered into the online submission form will not be published alongside your manuscript.

3. We note you have included a table to which you do not refer in the text of your manuscript. Please ensure that you refer to Table 1 and 2 in your text; if accepted, production will need this reference to link the reader to the Table.

Reviewers' comments:

Reviewer's Responses to Questions

**Comments to the Author**

1. Is the manuscript technically sound, and do the data support the conclusions?

Reviewer #1: Yes

Reviewer #2: Yes

2. Has the statistical analysis been performed appropriately and rigorously? 

Reviewer #1: Yes

Reviewer #2: Yes

3. Have the authors made all data underlying the findings in their manuscript fully available?

Reviewer #1: Yes

Reviewer #2: Yes

4. Is the manuscript presented in an intelligible fashion and written in standard English?

Reviewer #1: Yes

Reviewer #2: Yes

5. Review Comments to the Author

Reviewer #1: The paper is very informative on factors that hinder the health status and quality of life of indigenous people in Bangladesh. Many different issues have been raised in this study and of value for institutions and practitioners that deal with the improvement of living conditions, health care, and quality of life of indigenous people of tea gardens in Bangladesh.

The Introduction is offering a well-given overview of the problems of indigenous people in Bangladesh while Discussion is providing a critical review of the findings. The main disadvantages of this manuscript are related to text structure for which recommendations for improvement have been given. Comments and suggestions for the improvement of the manuscript can be found in the attachment.

Reviewer #2: First Considerations

This study explore the health status and quality of life of indigenous older people in Bangladesh remarking it significance due to the lack of a knowledge and evidence on this topic. Together with the huge gap in the health services and social supports for this population, being extremely old, male, presenting health illiteracy and not adopting health-seeking behaviors are variables that contribute to a poor quality of life

The title reflect properly the subject of the study, the abstract provide an accessible summary as well the keywords reflect accurately the content. The study is well written, structured in an organized, clear and systematized way.

Introduction: The research question is relevant and interesting once the topic add value to the subject area since there is a lack of a knowledge and evidence on the topic. The 1. Introduction sets out the argument and highlights the research related to the topic. Gives a clear idea of the target readership, as well the needed of it. The research aims are given.

Material and Methods: The methodology used is appropriate so it is the data analyses done. The design is suitable for answering the research question. The methods are well described, explained accurately how the data was collected, given in detail the procedures followed (e.g. ‘Sample size and sampling technique’ and ‘Study design and data collection’), which allows its replication and follow best practice.

However, in relation to:

. Quantitative data’s results, the presentation of the results of the instrument for assessing the quality of life used are not clear, specifically with regard to the relationship between the analysis the answers and the other sample variables.

. Qualitative data’s results, it remained to include the questions asked in the ten-in-depth interviews, as well how was the triangulation process of categorization (which steps were followed?). Methodologically it should also be referred the minimum and maximum duration (time) of the ten-in-depth interviews, and it should be given more than just one respondent’s answers to justify each categories found.

To mention that the research meets all applicable standards for the ethics of experimentation and research integrity.

Results and Discussion: The results and the discussion are described in simple terms, in a logical sequence. It evaluate the trends observed and explain the significance to wider understanding. A critical analysis of the data collected was done.

A positive point is the fact that are presented strengths and limitations of the study. Perhaps it would be more interesting to include this information in the topic of the Conclusion.

Conclusion: Consistent with the evidence and arguments presented, addressing the research question posed. It is suggested that this study can be an evidence base for future research since this population that has not been studied in these matters.

Data Tables an Figures: The included tables and figures aid to a more detailed reading of the results.

List of References: 65,9% of the reference papers are older than 5 years old (2015 backwards), and of these, 51,9% are prior to 2010. There are incomplete references [see, for example, 1, 10] as well some of them the year of the document is not indicated, only the date of consultation on the internet [see 3, 6, 7, 8, 9, 13, 32].

Final Considerations

This study presents the results of original research, making an interesting contribution to the field, which its adequately explained by the authors. It is presented sufficient evidence to substantiate the study claims and the interpretation of the data is equitable. The study is in line with the aims and scope of the Plos One. In this sense, the paper is PUBLISHABLE, with some revisions needed.

6. PLOS authors have the option to publish the peer review history of their article (what does this mean?). If published, this will include your full peer review and any attached files.

Reviewer #1: No

Reviewer #2: No

---

## [Author Response · Author response to Decision Letter 0]

26 Jan 2021

Response to Reviewers

I am grateful to both reviewers for their constructive reviews and helpful recommendations. I tried to address all the proposed revisions to the best of my capability. The detail response to the suggested revisions are given below:

Text structural revisions

All the suggested structural revisions by both reviewer-1 and reviewer-2 have been addressed. These include:

• Putting more than one references enclosed together in a single bracket.

• Exclusion of Table 1 and Table 2.

• Exclusion of headings like: ‘Quantitative findings’ and ‘Qualitative findings’.

• Placing all Tables and Figures first when mentioned and then placing the associated description/comments below them.

• Revising headings (delete colon, delete abbreviation)

• Change of headings for Table 4 and 5. (the headings have been changed as suggested)

• Completing references, (Ref no: 1,3,6,7,8,9,10,13,32).

Response to content revisions

Response to Reviewer-1

Section: Abstract

Comment 1 Page 2, Line: 25-27

‘Besides saying that goal of the study was looking into the health status and quality of life of indigenous people it would be good to mention that its determinants were investigated also.’

Response

The aim of the study is amended as follows:

‘This study explores the health status and quality of life along with their determinants among indigenous older people in Bangladesh in order to fill the knowledge and evidence gap on this topic.’

Section: Introduction

Comment 1: Page 6 line 120

Suggestion to write one clear sentence stating the aim of the study with mentioning about determinants.

Response: It has been amended as follows:

 ‘This study, therefore, aims to explore the health status and quality of life and to define its determinants among the indigenous older adults residing in tea gardens of Bangladesh.’

Comment 2: Page 6 line 121

Authors are talking in terms of impact between variables, however, this study only tested correlations. 

Response: This line has been amended as follows:

“‘However, the healthcare needs, health status, and its relationship with quality of life of indigenous older adults in Bangladesh are still unknown and unexplored, which signifies the need for research specific to this community.’’

Section: Methodology

Comment 1: Page 7 line 137

Suggesting to change current subheading (Sample size and sampling technique) into following one: "Sample and sampling technique"

Response: Subheading has been amended to “Sample and sampling technique”

Comment 2: Page 8 line 162

‘Study design and data collection section do not offer a detailed description of instruments used.’ Some suggestions were also offered to make this section more comprehensive.

Response: A separate section for instrument has been added with detail description of the questions used and their measurement technique. All the suggestions have been addressed. Can be found at Page-10, line-193 of the new manuscript.

The questionnaire has also been attached as Appendix-1.

Comment 3: ‘It would be good to provide the reliability that is, Cronbach's alpha for quality of life measure.’ 

Response: Cronbach's alpha value has been added- 

The Cronbach's alpha of the quality of life measurement scale was 0.66.

Comment 4: page 10 line 180

Suggested to add IDI guidelines in Appendix.

Response: The guideline used for conducting IDI has been added in Appendix-2.

Section: Results

Comment 1: page 12 line 200 to 209

It is proposed that text under the heading "Socio-demographic characteristic of subjects" along with table 3 and figure 1 are moved to the "Sample and sampling technique" section.

Response: Table-3 (now Table 1) and Fig 1 along with their associated texts have been moved to the "Sample and sampling technique" section.

Comment 2: Page 14 line 251

Suggestion to amend the line based on analysis of the manuscript.

Response: The line has been amended as per suggestion to-

"A large number (69.6%) of respondents reported difficulty in carrying out household tasks"

Comment 3: Page 19 line 332

Suggestion to add one more sentence providing the reader with a list of themes which emerged from qualitative analysis.

Response: The comment has been addressed as follows- ‘Three broad themes were emerged from the qualitative analysis- health scenario of the older adults, health-seeking behavior of the older adults, challenges faced by older adults in maintaining good health.’

Comment 4: page 15 line 279

‘it is suggested that this section begins by stating what was tested by logistic regression, what were its predictors and a criterion variable while defining its categories’

Response: A para has been added in response describing the variables used in logistic regression and what was measured by them. This can be found at Page 15 line 318 to 323 of the new/revised manuscript.

Comment 5: page 17 line 302

It was suggested that before explaining multiple linear regression result – ‘it would be informative to explain how independent variables were coded before the presentation of results in the table.’

Response: A para has been added in response explaining the coding of the independent variables. It can be found at page 17 line 349 to 357 of the new manuscript.

Comment 6: page 18 Table 5

Table 5 is not comprehensive enough. The column with variable names is too narrow. Try to fit variable names in one line.

Response: Table 5 (now Table 3) has been amended as suggested. Can be found at page 18 Table 3.

Comment 7: page 19 line 332

‘add one more sentence providing the reader with a list of themes which emerged from qualitative analysis.’ 

Response: It has been amended as follows: ‘The themes emerged from the qualitative analysis were common health problem of the indigenous older adults, practice of preventive health behavior, factors associated with health status and health-seeking behavior, inequality in health condition, challenges of older people in maintaining good health, and recommendation for improving health condition and health services to indigenous older adults.’

Section: Discussion

Comment 1: Page 24 line 470

Suggestion to reformulate the sentence- "These conditions have a substantial impact on daily life activities too with 70% of indigenous older people reporting difficulties with performing household tasks."

Response: The sentence has been reformulated as follows-

“Almost all (90%) the respondents were suffering from any kind of chronic health problem and a large number (70%) of them reported of having difficulties with performing household tasks. This indicates a possibility that the chronic condition might have an influence in limiting their daily activities, which can be investigated with further studies.”

Comment 2: Page 27 line 539

It is suggested that a couple of words are shared regarding the practical implications of this study.

Response: It has been addressed by adding the following sentence-

“This study implies the need for policies to improve the coverage of the old-age welfare system, including marginalized groups such as indigenous peoples, and to develop tailored health funding and awareness-raising strategies for indigenous older adults.

health strategies for older population etc.”

Comment 3: Figure-5: The explanation for abbreviations used in Figure 5 is missing in order all information provided in it is tangible.

Response: Full form of the abbreviations have been added.

Response to Reviewer 2:

Comment 1: 

Quantitative data’s results, the presentation of the results of the instrument for assessing the quality of life used are not clear, specifically with regard to the relationship between the analysis the answers and the other sample variables.

Response: 

To increase the clarity of the instrument and analysis for assessing quality of life, a description of multiple linear regression analysis exploring the relationship between quality of life and predictor variables (socio-demographic & chronic condition) is added into the result section before the table, explaining how the variables were coded and a clarification of adjusted and unadjusted models. The Cronbach alpha score of the quality of life score scale is also added.

Comment 2:

Qualitative data’s results, it remained to include the questions asked in the ten-in-depth interviews, as well how was the triangulation process of categorization (which steps were followed?). Methodologically it should also be referred the minimum and maximum duration (time) of the ten-in-depth interviews, and it should be given more than just one respondent’s answers to justify each categories found.

To mention that the research meets all applicable standards for the ethics of experimentation and research integrity.

Response:

The IDI guideline has been attached as Appendix 2. 

Couple of lines have been added mentioning data triangulation and duration of the interviews- IDI participants were selected from health service providers of different educational and social background as part of the data triangulation process. The respondent were 3 registered medical doctors, 3 compounders, 3 paramedics, and one community healthcare worker. Each interview lasted from 30 to 60 minutes. Can be found at page 9 line 184-188 of new manuscript. 

Another line has been added mentioning triangulation of qualitative and quantitative findings at page-12 line 252-254. ‘The qualitative data were used to supplement the quantitative findings and triangulated interpretations were discussed to demonstrate a comprehensive health scenario of the indigenous older adults.’

Ethics statement has been added to the methodology section- ‘Informed written consent was taken from all the respondents. Anonymity and confidentiality of the respondents were maintained throughout the data collection, handling, and analysis process. Ethical approval was taken from Institutional Ethical Review Board of CIPRB.’ (page 10 line 189-191)

---

## [Decision Letter · Decision Letter 1]

17 Feb 2021

Health and Well being of Indigenous Older Adults Living in the Tea Gardens of Bangladesh

PONE-D-20-33976R1

Dear Dr. Rahman,

We’re pleased to inform you that your manuscript has been judged scientifically suitable for publication and will be formally accepted for publication once it meets all outstanding technical requirements.

Kind regards,

Filipe Prazeres, MD, MSc, Ph.D.

Academic Editor

PLOS ONE

Additional Editor Comments (optional):

Reviewers' comments:

Reviewer's Responses to Questions

**Comments to the Author**

1. If the authors have adequately addressed your comments raised in a previous round of review and you feel that this manuscript is now acceptable for publication, you may indicate that here to bypass the “Comments to the Author” section, enter your conflict of interest statement in the “Confidential to Editor” section, and submit your "Accept" recommendation.

Reviewer #1: All comments have been addressed

Reviewer #2: All comments have been addressed

2. Is the manuscript technically sound, and do the data support the conclusions?

Reviewer #1: Yes

Reviewer #2: Yes

3. Has the statistical analysis been performed appropriately and rigorously? 

Reviewer #1: Yes

Reviewer #2: Yes

4. Have the authors made all data underlying the findings in their manuscript fully available?

Reviewer #1: Yes

Reviewer #2: Yes

5. Is the manuscript presented in an intelligible fashion and written in standard English?

Reviewer #1: Yes

Reviewer #2: Yes

6. Review Comments to the Author

Reviewer #1: The authors have successfully addressed all the suggestions for improvements previously provided. As a result, the manuscript is clear, coherent while data support conclusions. Therefore, the manuscript is ready for publication.

Reviewer #2: Due to the (very cordial) response of the author, (almost) all of the recommendations were taken into account by him. In that sense, I believe that this manuscript ca be accepted for publication.

7. PLOS authors have the option to publish the peer review history of their article (what does this mean?). If published, this will include your full peer review and any attached files.

Reviewer #1: No

Reviewer #2: No

---

## [Editor Report · Acceptance letter]

23 Feb 2021

PONE-D-20-33976R1 

Health and Wellbeing of Indigenous Older Adults Living in the Tea Gardens of Bangladesh 

Dear Dr. Rahman:

I'm pleased to inform you that your manuscript has been deemed suitable for publication in PLOS ONE. Congratulations! Your manuscript is now with our production department. 

Kind regards, 

on behalf of

Prof. Filipe Prazeres 

Academic Editor

PLOS ONE